# Underestimation of strong wind speeds offshore in ERA5: evidence, discussion, and correction

Rémi Gandoin[1], Jorge Garza[1]

[1]C2Wind ApS, Fredericia, 7000, Denmark.

*Correspondence to*: R. Gandoin (rga@c2wind.com)

**Abstract.** Offshore Wind power plants have become an important element of the European electrical grid. Studies of metocean site conditions (wind, sea state, currents, water levels) form a key input to the design of these large infrastructure projects. Such studies heavily rely on reanalysis datasets which provide decades-long model time series over large areas. In turn, these time series are used for assessing wind, water levels and wave conditions, and are thereby key inputs to design activities such

as calculations of fatigue- and extreme loads, and platform elevations. In this article, we address a known deficiency of one these reanalysis datasets, ERA5, namely that it underestimates strong wind speeds offshore. If left uncorrected, this poses a design risk (large- and extreme wind, waves and water level conditions are underestimated). Firstly, comparisons are made against CFSR/CFSv2 reanalyses as well as high quality wind energy specific in-situ measurements from floating LiDAR systems. Then, the ERA5 surface drag formulation and its sea state dependency are analysed in detail, the conditions of the

bias identified, and a correction method is suggested. The article concludes with proposing practical and simple ways to incorporate publicly available, high-quality wind energy measurement datasets in air-sea interaction studies alongside legacy measurements such as met buoys.

## 1 Introduction

Offshore wind power plants help reduce carbon emissions from the power sector. They have gradually evolved from small

demonstration projects (Vindeby, commissioned in 1991) to commercial-scale demonstration projects (Horns Reef 1, Nysted) in the early 2000s. Today, they stand as integral components of the European power grid (ENTSO-E, 2024).

Atmospheric Boundary Layer (ABL) wind datasets from Numerical Weather Prediction modelling systems (NWP models) are routinely used for the purpose of assessing the offshore wind resource and for characterising sea state, water level and current conditions at offshore wind farm locations (NWP models provide inputs to hydrodynamic- and spectral wave models). Global

reanalysis datasets such as CFSR (Saha et al., 2010), CFSv2 (Saha et al., 2014) and ERA5 (Hersbach et al., 2020) are widely used for these purposes as they are publicly available, free of charge, and cover long periods (decades).

Despite performing among the best (see (Ramon et al., 2019) , a study considering 77 tall tower sites which concludes that "ERA5 near-surface wind dataset offers the best estimates of mean wind speed and variability at turbine hub heights"), the NWP model used for producing ERA5 suffers from a major drawback for engineering purposes: it underestimates strong wind

speeds offshore close to the surface, for instance at 10 m (a nominal elevation often used for hydrodynamic- and spectral wave modelling). This is documented for instance in (Bentamy et al., 2021), a study which uses in-situ far- and near offshore measurements, see its figures 4 and 5 which display an ERA5 bias for strong wind conditions. Similarly, (Alday et al., 2021) refer to (Pineau-Guillou et al., 2018) for documenting the ERA5 bias and propose a piece-wise linear correction. The bias is documented as well in (Solbreke et al., 2021; Spangehl et al., 2023; Meyer et al., 2023) at measurement locations in the North

Sea where ERA5 performs worse than other datasets. Therefore, for engineering applications the ERA5 10 m wind speed values are often corrected to ensure that site conditions design values are not underestimated, see for instance (DHI, 2023). In effect, for design, slightly conservative values are typically desirable: that is: model results that underestimate large wind speeds, and thereby also large waves, pose a design risk (of too small loads, and also too low platform elevations). As a result, and despite their shortcomings (differences in land/sea masks and grid resolution, poorer correlation with in-situ

measurements, no wind speed time series close to a modern turbine hub height), CFSR and CFSv2 remain, in the authors' experience, the preferred datasets for driving engineering hydrodynamic- and spectral wave modelling systems.

This ERA5 bias has not been widely discussed in the scientific literature, and it may not be clear to all ERA5 users that the data need to be corrected. Also, the methods published so far only partially address the bias: most often, they correct 10 m winds only, and/or use site-specific corrections; see (Alday et al., 2021) or (DHI, 2023). The present article proposes a novel

approach to both topics. After having provided elements of wind speed modelling in Sect. 1, we compare ERA5 and CFSv2 model time series at selected locations, with each other and against in-situ measurements in Europe and America in Sect. 2. The ERA5 strong wind bias is discussed in Sect. 3: a detailed analysis of the ERA5 drag formulation is provided and a simple correction method is suggested for wind speeds between 10 and 100 m using analytical ABL wind profile expressions from the literature. The measurement datasets all come from high-quality, publicly available met mast and Floating LiDAR System

(FLS) datasets; these are described in Sect. 1.

The main objectives of this article are 1) presenting evidence of the ERA5 underestimation of strong wind speeds, 2) discuss the reason for this underestimation, and propose a simple corrective method and 3) argue for using publicly available, high quality FLS measurements and met mast datasets for air-sea interaction studies, along with legacy measurements such as met buoys. In Sect. 4, and with references to the recent literature, suggestions are made regarding practical actions and research

initiative.

This article provides references to recent works regarding air-sea interaction and drag formulations. Yet, it does not take a scientific stand on the nature of these interactions. Instead, it merely tries to bring a practitioner's perspective to this long-lasting discussion, that is: for design purposes, an accurate depiction of both wind *and* sea state in reanalysis datasets is required, and useful, quality datasets are readily available for validation work.

## 1.1 Elements of wind profile modelling

As explained for instance in (Peña et al., 2008a) and its references such as (Stull, 1988), in the layer close to the surface where the Monin-Obukhov Similarity Theory (MOST) is valid, the mean wind speed $U$ at a given elevation $z$ above the surface is given by:

$$U(z) = \frac{u_{*0}}{\kappa}\left(\ln\left(\frac{z}{z_0}\right) - \psi_m\left(\frac{z}{L}\right)\right)$$ (1)

where $u_{*0}$ is the friction velocity at the surface, $z_0$ is the roughness length, $\kappa$ is the Von Karman constant (here taken equal to 0.4), $L$ is the Obukhov length and $\psi_m$ is an atmospheric stability-dependent function derived from experiments. Above the surface layer, this expression needs to be supplemented with additional terms: the height of the boundary layer $z_i$ and a length scale $L_{MBL}$ which is a characteristic length scale of the eddies in the ABL, see originally (Gryning et al, 2007) and its references.

Over water, the Charnock relationship is used for linking roughness length and friction velocity, see (Peña et al., 2008) and Eq. (3.26) of (ECMWF, 2016a):

$$z_0 = \alpha_{\text{Ch}} \cdot \frac{u_{*0}^2}{g} + \alpha_{\text{M}} \cdot \frac{\nu}{u_{*0}}$$ (2)

where $\alpha_{\text{Ch}}$ and $\alpha_{\text{M}}$ are sea state-dependent parameters and $\nu$ is the air kinematic viscosity (term only relevant for very small wind speeds, we use $\alpha_{\text{M}} = 0.11$ following section 3.2.4 of (ECMWF, 2016a)). Formulations (1) and (2) are widely used in NWP modelling systems such as the Global Forecast System (GFS), the Integrated Forecast System (IFS) or the Weather Research and Forecasting (WRF) Model. The term $\alpha_{\text{Ch}}$ is referred to as the Charnock parameter and is either kept constant (for instance in (Peña et al., 2008) or in CFSR/CFSv2, see (Renfrew et al., 2002)) or made dependent on sea state conditions. In this article, we focus on the IFS Cy41r2 (ERA5) implementation, see Eq. (3.11) of (ECMWF, 2016b), where the atmospheric- and ocean models are coupled via:

$$\alpha_{\text{Ch}} = \frac{\hat{\alpha}}{\sqrt{1 - \frac{\tau_{\text{w}}}{\tau}}}$$ (3)

where $\tau$ is the wind stress ($\rho_a u_{*0}^2$ where $\rho_a$ is the air density), $\tau_w$ is the wave stress (from the waves to the atmosphere) and $\hat{\alpha}$ is a constant. For all practical purposes, a neutral drag coefficient $C_{\text{d,n}}$ can be derived, and often used for comparing model- and measurement results:

$$C_{d,n}^2 = \frac{u_{*0}}{U_n} = \frac{\kappa}{\ln\left(\frac{z}{z_0}\right)}$$

(4)

Where $U_n$ is the wind speed for neutral conditions evaluated using Eq. (1) and $\psi_m(z/\infty) = \psi_m(0) = 0$ (negligible buoyancy).

## 1.2 Measurement data description

In this article, we use well documented and validated, high-quality, publicly available measurement datasets from the wind energy industry such as Floating LiDAR Systems and a met mast (cup anemometer); see Figure 1. Legacy instruments such a
met buoys have been left out intentionally due to the poor quality and traceability of these measurements in comparison with the former datasets. A discussion is provided in Sect. 4 on future works and the advantages of adding such wind energy measurements alongside legacy instruments to decrease modelling uncertainty.

The measurements have been chosen from the comprehensive list of datasets available on the Wind Resource Assessment Group wiki page[1], their locations are marked in red in Fig. 2 and a high-level description is provided in Table 1 (except for
M6 and 62001, where no measurement data are used). All measurement locations lie far offshore, where land/sea mask effects are negligible for the wind directional bins selected for the analyses (see Sect. 2.3).

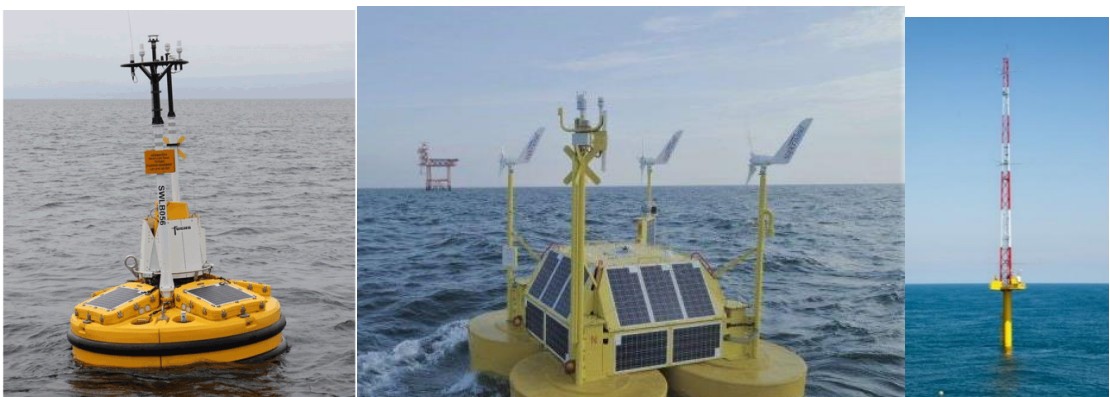

**Figure 1: Photographs of the Floating LiDAR Systems (left: Fugro, middle: Eolos) as well as the IJmuiden mast (right). Sources:**
**Fugro, Eolos, Wind op Zee.**

---

[1] See https://groups.io/g/wrag/wiki/13236.

**Table 1: High-level description of the measurement datasets used in this article. Detailed information can be found in the references provided in the table.**

| ID | Lon °E | Lat °N | Type | Period | Elevations [mASL] | Source | References |
|---|---|---|---|---|---|---|---|
| IJM | 3.436 | 52.998 | Mast (cups) | Nov11-Mar16 | {26; 57; 91} | Wind op Zee | (Quaeghebeur and Zaaijer, 2020) |
| TNWA | 5.551 | 54.018 | FLS | Jun19-Jun20 | {4; 30; …; 250} | RVO | (Fugro, 2022) |
| E05 | -72.715 | 39.969 | FLS | Aug19-Sep21 | {20; …; 200} | NYSERDA | (EOLOS, 2020) |
| E06 | -73.429 | 39.546 | FLS | Sep19-Mar22 | {20; …; 200} | NYSERDA | (EOLOS, 2020) |
| Lot 1 | 6.301 | 56.628 | FLS | Nov21-Nov23 | {4; 30; …; 270} | ENS | (Fugro, 2023) |
| Lot 2 | 6.457 | 56.344 | FLS | Nov21-Nov23 | {4; 30; …; 270} | ENS | (Fugro, 2023) |


All FLS measurements have been validated as per the Carbon Trust Offshore Wind Accelerator Roadmap Stage 3 (Carbon Trust, 2018). That is: both types of LiDAR and FLS have been validated dozens of times against reference measurements (cups, or LiDAR validated against cups), and these tests have repeatedly shown mean relative deviations smaller than 2%. Examples of validations are provided in Fig. A1 and A2 in Appendix A. Large number of publicly available validation reports

have been collected by the authors, see the supplementary material to this paper. For the specific case of the Fugro FLS, from the RVO campaigns[2] 16 validation reports are available together with additional studies such as (Kelberlau, 2022) and (Kelberlau and Mann, 2022) showing similarly very small deviations against several cup anemometer measurements offshore. For the EOLOS FLS, three validation reports are available in the above-mentioned online repository, and (Araújo da Silva, 2022) provides a thorough description and validation of the device at the IJmuiden met mast.


---

[2] See https://offshorewind.rvo.nl/

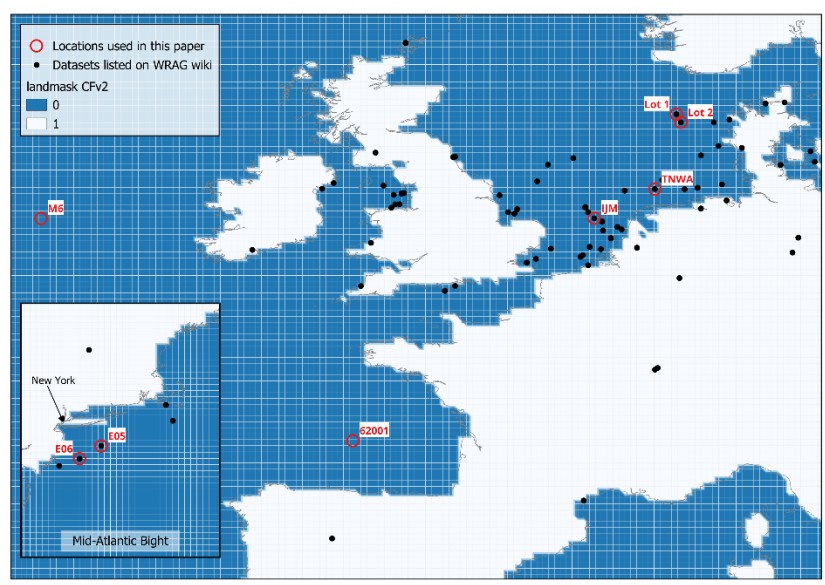

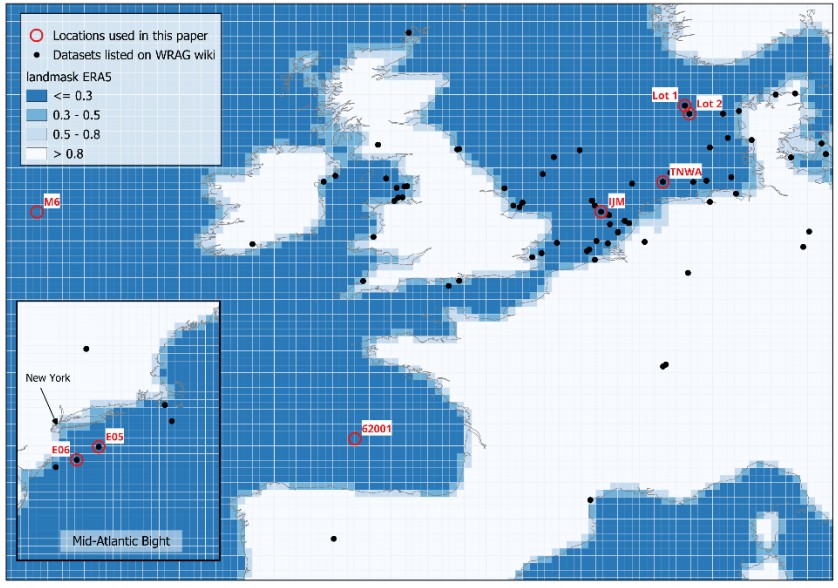

**Figure 2:** Location of the publicly available wind energy specific wind datasets (black dots), together with the analysis locations used for this paper. The small box on the bottom-left of the subfigures shows the Mid-Atlantic Bight off the USA East Coast, while the larger map shows the North Sea. The two maps show land/sea masks for CFSv2 (top) and ERA5 (bottom). For ERA5, and for IFS in general, land/sea mask values range from 0 to 1 and indicate[3] the fraction of land in the model cell.

---

[3] See https://confluence.ecmwf.int/display/FUG/Section+2.1.3.1+Land-Sea+Mask#Section2.1.3.1LandSeaMask-Land-Seamask.

All FLS used in this study are equipped with ZX Lidars continuous wave LiDARs; see (Knoop et al., 2021) for a research validation study. The 10-minute data have been filtered using a regular filter based on the average number of valid samples

during one scan (so-called minimum number of packets); it is here set to 18. Fugro uses a threshold of 9 (see Table 3.2 of (Fugro, 2021)), and validation studies such as (TNO, 2021) show that for this type of LiDAR the accuracy of the measurement is not significantly sensitive to the value of the threshold. A thorough and quantitative overview of availability statistics for these instruments are provided in the references stated in Table 1. For comparisons with model data, the data have been hourly averaged and only time periods with at least 5 valid 10-minute timestamps data have been used.

**1.3 Derivation of a 10m wind time series from measurements**

As explained later in Sect. 3.2, when $u_{*0}$, $L$, and $\alpha_{Ch}$ are known, wind speed at any elevation in the surface layer can be computed (an example is given for the 26.1 m cup anemometers at IJM). Therefore, deriving a 10 m wind from measurement data is not always necessary,

However, for practical reasons this often needs to be done (CFSR and CFSv2 winds are available at 10 m, or, traditionally for

spectral wave modelling). In the present article, two methods have been considered for every 10-minute timestamp: 1) interpolating using a power law the measurements between the 4 m sonic anemometers and the lowest LiDAR measurement elevation, and 2) fitting a power- or log law to the LiDAR measurements up to- and including 80 m, and then extrapolating down to 10 m. Both methods add some uncertainty to the derived 10 m value: for 1), the uncertainty mainly lies in the uncertainty of the 4m sonic anemometers; for 2), and in particular for stable conditions, the wind profile may not be well fitted

by a power law. To alleviate these uncertainties, the present study focuses on wind speeds larger than 15 and 20 m/s (where stable conditions are very rare; this was checked from both reanalysis data time series but also the literature, see (Sathe et al., 2011) for the North Sea) and in Sect. 2.3 the comparison is made for unstable- and neutral conditions only by limiting the range of air-sea temperature difference to $\Delta\theta = T_{4m} - SST < [-2; 0.5[[\square]$ °C (North Sea) and $\Delta\theta = T_{4m} - SST < 0.5$°C (Atlantic Bight). The reason for choosing two different ranges of temperature differences, is that in the Mid-Atlantic Bight

strong wind conditions occur during winter for very unstable conditions. For all these comparisons, the mean wind speed measurement profile is provided to check the validity of the extrapolation method. Furthermore, the uncertainty of the extrapolation method has been verified using proprietary Fugro FLS measurements located in Northern Scotland where 12 m LiDAR measurements are available (the exact location is confidential): see Fig. 3 which shows that the errors are the smallest when considering method 1). This method has thereby been chosen in Sect. 2.3, but it has been checked that the conclusions

of the analysis are not sensitive to the choice of the method (i.e., that the ERA5 10 m wind speeds are smaller than measured values, also when considering measurement uncertainty).

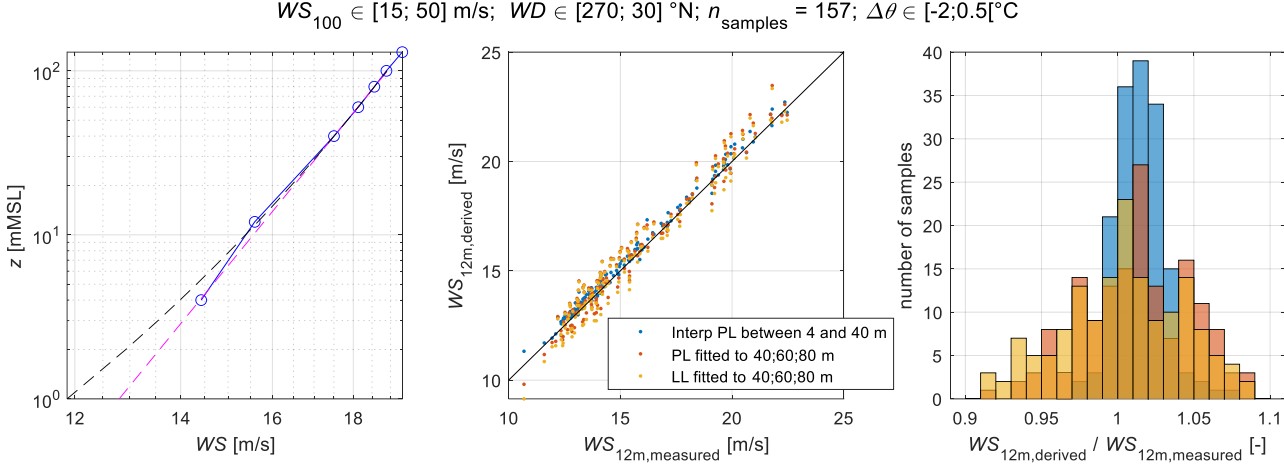

**Figure 3: The middle- and right-hand side panels figure show comparisons of 12 m hourly Wind Speed (*WS*) measurements from a Fugro FLS (undisclosed location) with 12 m time series derived using the three methods discussed in the text and stated in the legend (PL stands for Power Law and LL stands for Log Law). The left-hand side panel shows the corresponding mean wind speed profile, where the blue markers are FLS measurements, the black line is the fitted log law and the magenta line the fitted power law. Measurements have been filtered for 100 m wind speeds (*WS*$_{100}$) between 15 and 50 m/s, as well as for wind directions (*WD*) between 270° and 30°.**

## 1.4 Model data

For this study, CFSR (Saha et al., 2010) and CFSv2 (Saha et al., 2014) data have been downloaded using DHI's MetOcean On Demand (MOOD) web interface[4]. Data from ERA5 (Hersbach et al., 2020) have been downloaded from MOOD as well for the locations M6, Lot 1, Lot 2, E05 and E06. For the locations TNWA, IJM and 62001 the ERA5 data was also fetched from the Copernicus Data Store (CDS), and for these locations it was verified that both sources are identical. When comparing with in-situ measurements, for IJM and TNWA the data was interpolated at the measurement location. For E05, E06, Lot1 and Lot2 the nearest node was used. The following parameters have been used (all hourly time series):

➤ For CFSR/CFSv2: 10 m wind speed and direction, air- and sea surface temperature.

➤ For ERA5:

 ○ From Metocean on Demand: 10 and 100 m wind speed and direction.

 ○ From the CDS: same as above plus 2 m air temperature, sea surface temperature, friction velocity, roughness length, Charnock coefficient, sensible heat flux, dew point temperature, pressure at the sea surface.

---

[4] See https://www.metocean-on-demand.com/, where it is stated that "*The data is extracted as discrete (non-interpolated) values of the model grid cell.*"

## 2 Comparisons of ERA5 and CFSR / CFSv2 with measurements

In the literature, multiple comparisons between in-situ measurements and IFS model wind speed close the surface have concluded that for strong wind conditions the IFS model wind speeds are smaller than measured values and smaller than other reanalysis datasets (global, or regional), see for instance (Fery et al., 2018) for ERA Interim and (Bentamy et al., 2021), (Alday et al 2021), (Solbreke et al., 2021), (Spangehl et al., 2023), (Meyer et al., 2023) for ERA5.

### 2.1 Comparisons between ERA5 and CFSv2

Examples of differences with ERA5 (IFS) and CFSv2 (GFS) model results are provided below for two locations: the M6 buoy off the west coast of Ireland, and the TNW FLS; see Fig. 4 Similar trends are visible across several locations in the northwest European shelf, see the supplementary material. Figure 5 shows that at the 62001 buoy location and when separating the dataset between short- and long fetches, the relative difference between the models seems to be larger for short fetches; as discussed in Sect. 3 this is the sign that the difference between the models is driven by the dependency of the ERA5 drag formulation to the sea state. In this example, short fetches are defined as wind directions where wind comes from land across the Bay of Biscay while wind direction oriented towards the Atlantic Ocean are considered long fetches, see Fig. 2. This effect is of the same magnitude at all locations when considering the CFSR data (1979-01-01 to 2011-04-01) which have a coarser resolution than ERA5, see the supplementary material provided with this paper.

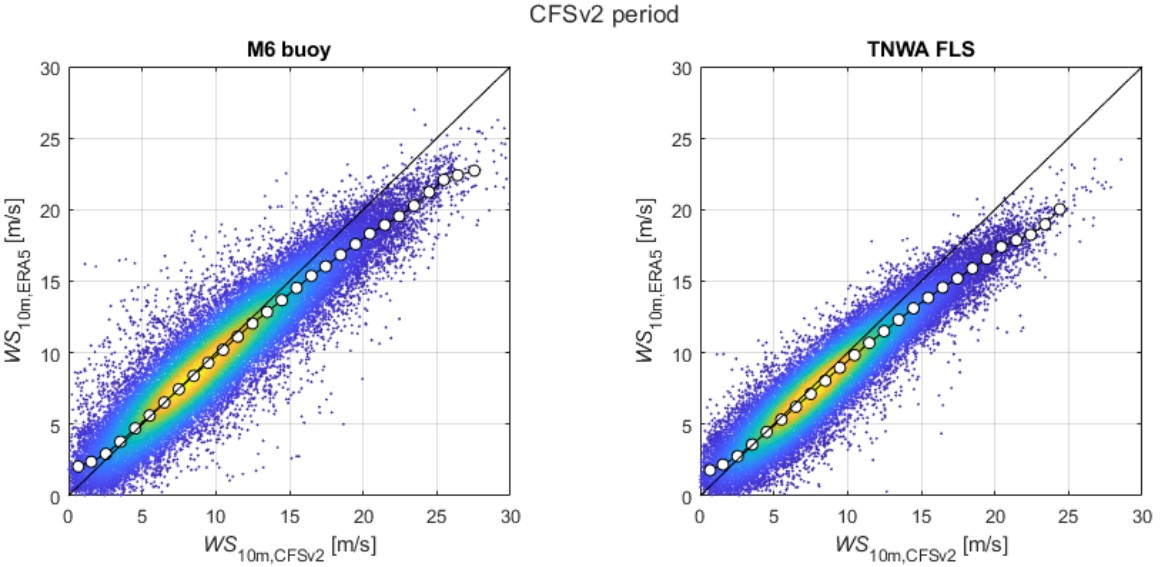

**Figure 4: For two locations (see Fig. 2), comparison of the ERA5 and CFSv2 10m wind speeds. For wind speeds above approximately 10 m/s, the ERA5 values are smaller than the CFSv2 values; this effect is stronger at TNWA than at M6. The density of the scatter plot uses a colormap, from blue (low density, few points) to yellow (high density, many points). The white dots at the binned mean values (for bins with more than 10 points).**

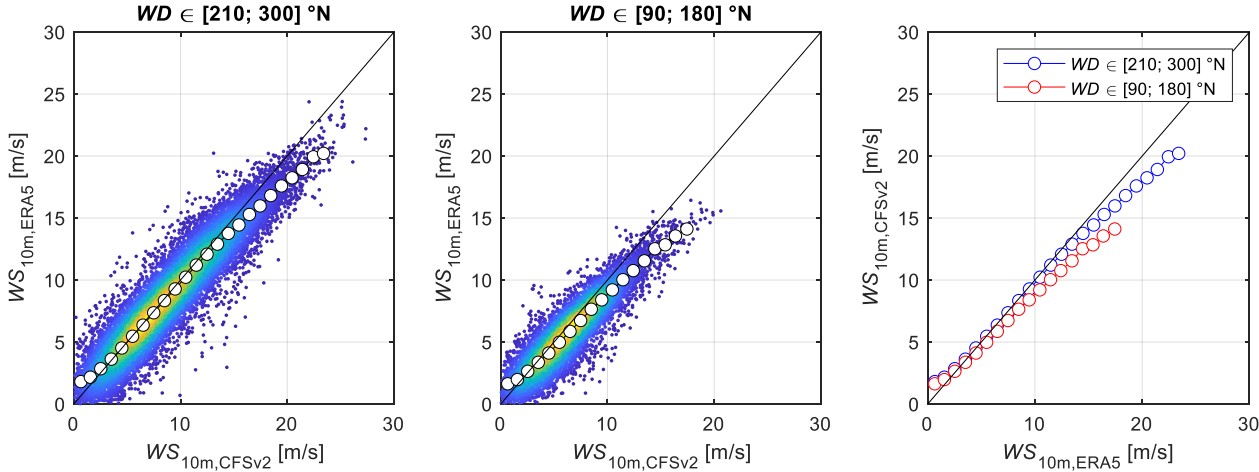

**Figure 5: As for Fig. 4, this figure shows a comparison of ERA5 and CFSv2 10m wind speeds, this time at the 62001 buoy and for two wind directional bins corresponding to very long fetch and short fetch. The difference between the two model results is larger for short fetches. The white dots at the binned mean values (for bins with more than 10 points).**

## 2.2 Comparisons between ERA5 and measurements

Using the method explained in Sect. 1.3, 10 m wind speed time series have been derived from FLS measurements. These values compared with ERA5 and CFSv2 model data for 5 measurement locations in Fig. 6 (Lot 1) and in Appendix A in Fig. A3 to A6. For all locations, the ERA5 values are smaller than the measured values, and smaller than the CFSv2 values.

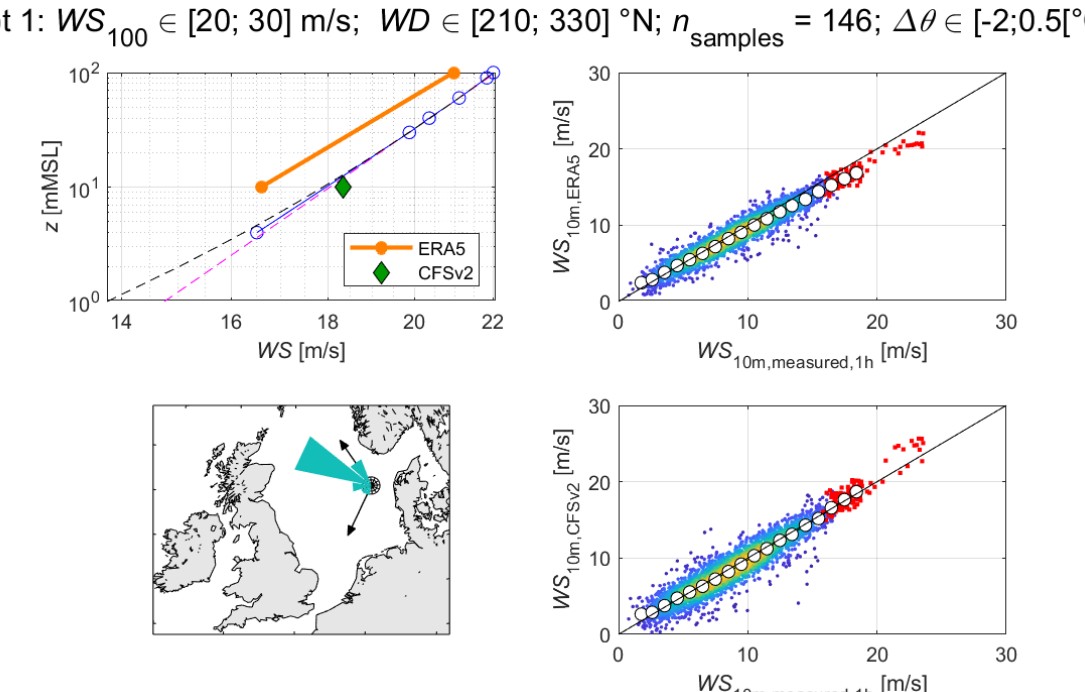

**Figure 6: For one FLS location in the Danish North Sea, comparison between modelled and measured hourly mean wind speeds. The white dots at the binned mean values (for bins with more than 10 points). A subset of large wind speeds (selected using the FLS measurements at 100m), are highlighted in red. The plot to the top-left shows mean wind speed profiles for this subset, where the blue line represents measurements. The dashed black line is a power law fit to the measurement data below 80m, the magenta line represents a log law. The 10m measured data on the x-axis of the scatter plots has been interpolated between the 4m sonic anemometer and the lowest LiDAR elevation; it can be checked on the top left that this is a valid approach which introduce an uncertainty smaller than the difference between the two models. A wind rose corresponding to the subset of large wind speeds (in red on the scatter plots) is shown on the map.**

## 3 Addressing and correcting the ERA5 drag formulation

In Sect. 3.1, using ERA5 data downloaded from the CDS, we check that ERA5 wind speeds can be derived from friction velocity, Charnock parameter and the Obukhov length; then we analyze the behavior of the ERA5 drag formulation for different sea state conditions and conclude that the ERA5 bias likely comes from the asymptotic behavior of Eq. (3) for large values of $\tau_w/\tau$. Then, in Sect. 3.2 we propose a simple correction which consists of capping, or keeping constant, the values of the Charnock parameter.

## 3.1 Drag formulation in ERA5

As explained in Sect. 1, friction velocity, roughness length and mean wind speed are interlinked via Eq. (1) and (2). On one hand, the ERA5 analysis 10 m wind speed time series is readily available (via its two horizontal components) on the CDS. On the other hand, friction velocity $u_{*0}$ and roughness length $z_0$ are only provided as forecast values. Please note that this forecast

roughness length time series is faulty and should not be used[5]; instead, the best way to estimate $z_0$ from CDS data is to use the Charnock coefficient time series (available from the CDS) and $u_{*0}$ as in Eq. (2). As a result, ERA5 10 m wind speeds time series can be reconstructed solely using Eq. (1) with $u_{*0}$, $\alpha_{Ch}$ and $L$ computed from CDS data[6]: as shown in Fig. 7 this gives satisfactory results. Furthermore, the formulations presented in (Gryning and al., 2007) can be used for deriving very realistic

100 m wind speeds, see Fig. 7 as well (this is helpful to practitioners who may want to derive wind speeds at larger elevations). As explained above, $u_{*0}$ is provided as a forecast value while the 10 m wind speed from the CDS comes from the analysis (ERA5 contains both analysis and forecast fields); also, the Obukhov length $L$ is computed from other model fields, and does not necessary correspond, for every timestamp, to the value used in the IFS. These differences explain the scatter between the reconstructed 10m wind speeds and the ones from the CDS, on the second plot in Fig. 7.

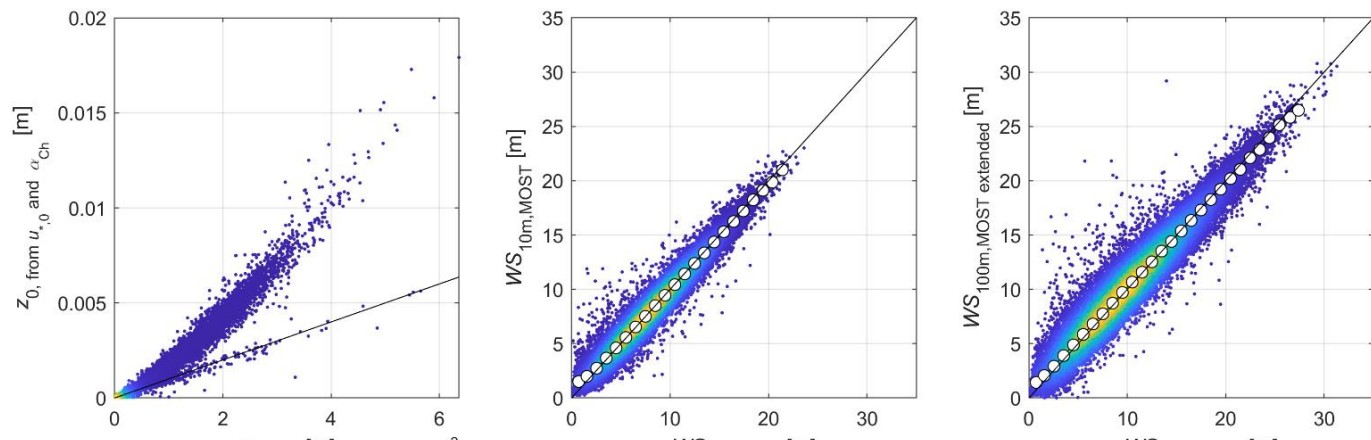

**Figure 7: For ERA5 at the TNWA location, the left-hand side of this figure shows comparisons between the roughness length values from the CDS (faulty) and the ones computed using the friction velocity and the Charnock parameter time series. The plot in the middle shows a comparison between the 10m wind speed from the CDS and the one derived using MOST with the friction velocity,**
**the roughness length derived from friction velocity and Charnock parameter, and the Obukhov length computed from CDS time series. The plot to the right shows a comparison between the 100m wind speed time series from the CDS and the one derived from the same parameters as for the middle plot, plus the surface layer extension model from (Gryning et al, 2007). The white dots at the binned mean values (for bins with more than 10 points). Overall, this figure shows that the wind speed values from the CDS at 10 and 100m can easily be reproduced using well known wind profile expressions.**

---

[5] Item 18 on https://confluence.ecmwf.int/display/CKB/ERA5%3A+data+documentation#ERA5:datadocumentation-Knownissues.

[6] For the present study, we have used $u_{*0}$, and the sensible heat flux, the air- and dew point temperature at 2 m as well as the surface pressure. An alternative method is described on the ECMWF user support website at https://confluence.ecmwf.int/display/CKB/ERA5%3A+How+to+calculate+Obukhov+Length.

## 3.2 Bias explanation and correction

From the above discussion, the results of the ERA5 drag formulation can be analysed further, with regards to Eq. (3) and the sea state dependency of the Charnock parameter. For increasing values of $\tau_w/\tau$, the resulting value of $z_0$ has been computed for a wide range of 10 m wind speeds using Eq. (1) (2) iteratively, and the results are shown against ERA5 time series at

TNWA and the 62001 buoy in Fig. 8. For strong wind speeds the $z_0$ values are relatively larger for the former than for the latter, and for large values of $\tau_w/\tau$, small changes of this ratio lead to very large changes in $z_0$: for instance, at 15 m/s, a change from 0.99 to 0.995 in $\tau_w/\tau$ leads to a 60% increase in $z_0$. Since, as shown in Fig. 9, the ERA5 and CFSR/CFSv2-derived friction velocity values are similar, this increase in roughness length leads to a decrease in mean wind speed.

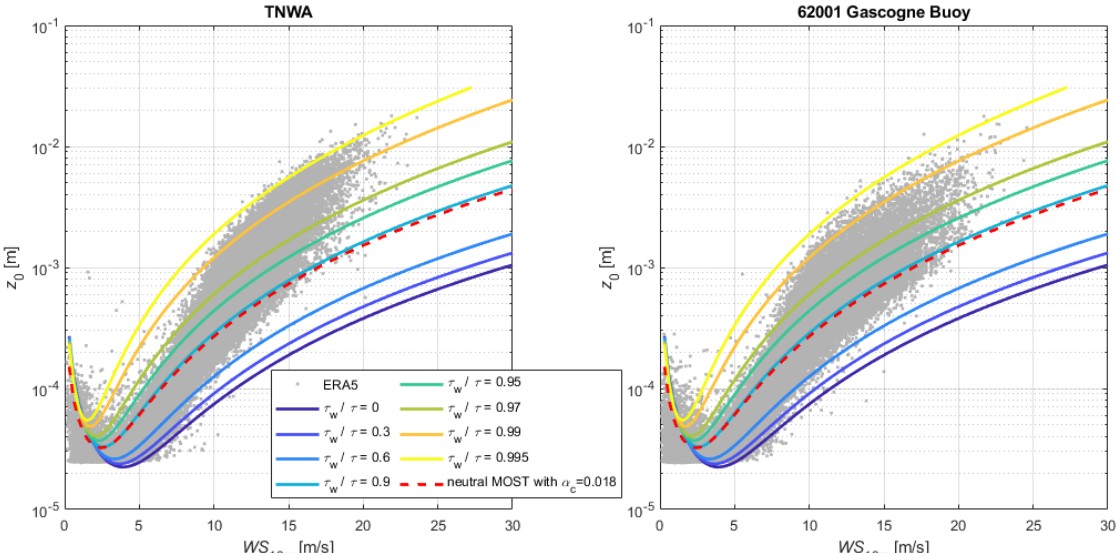

**Figure 8: For two locations, this figure shows a scatter plot of ERA5 roughness lengths values against 10m wind speed. Using an iterative process, Eq. (1), (2) and (3) have been solved for different values of ratios between wave- and wind stress, and for a constant Charnock parameter value of 0.018; the results are shown on the figures.**

In IFS, as $\tau_w/\tau$ increases, the Charnock coefficient increases; Fig. 10 shows that this is particularly the case for short fetch conditions in relatively shallow waters like the North Sea. For such conditions and for a given value of significant wave height,

the peak period of the spectra is typically smaller than for fully developed, long fetch conditions. The bias in IFS may then be related to the growth rate parameterisation described in Sect. 2.2 of (ECWMF, 2016b). This should be investigated in future works, for instance in ERA6 and ERA7 pre-production validation studies; see Sect. 4.

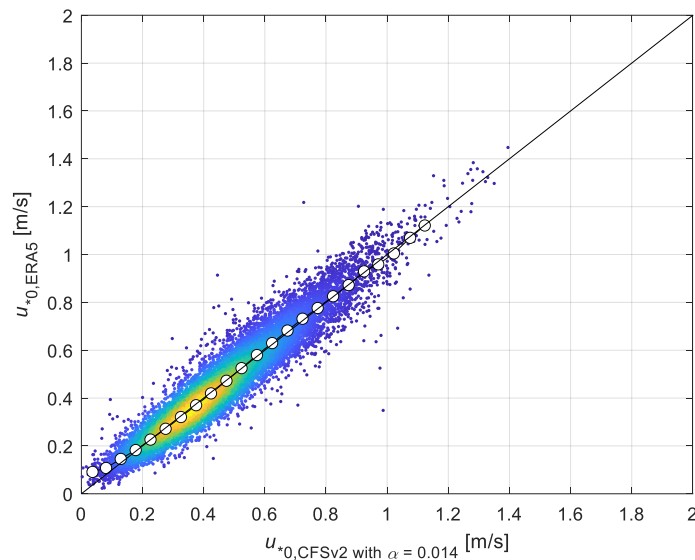

**Figure 9: For the TNWA FLS location and for neutral conditions, this figure shows a comparison between friction velocities values from the ERA5, and the ones derived iteratively using Eq. (1) and (2) and the 10 m CFSv2 wind speed time series (with a constant Charnock coefficient of 0.014, see (Renfrew et al., 2002). The white dots at the binned mean values (for bins with more than 10 points).**

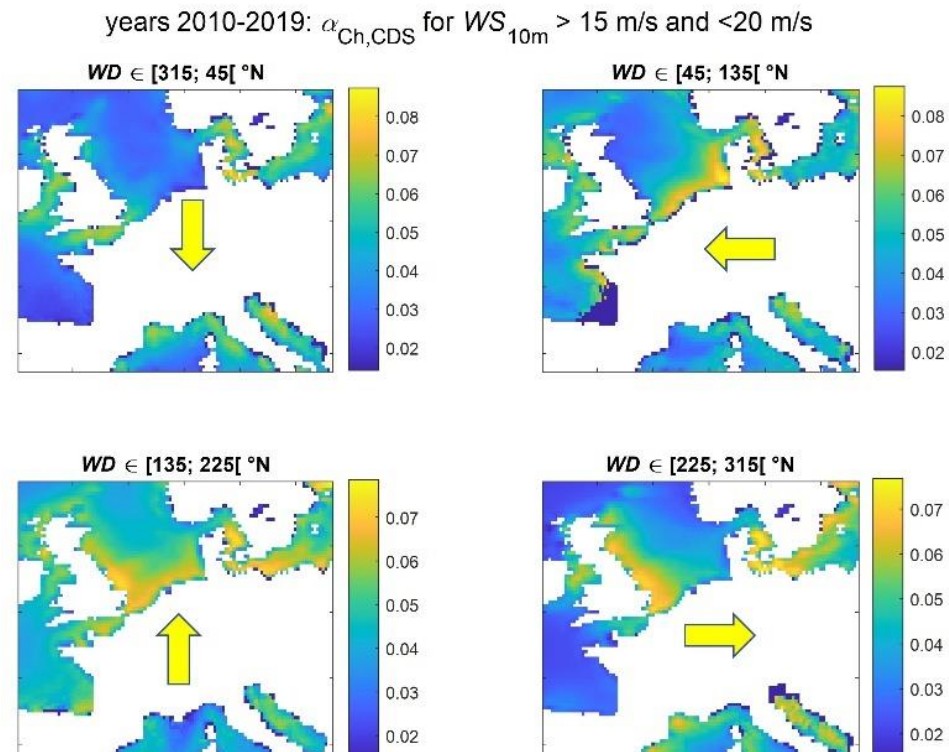

years 2010-2019: $\alpha_{Ch,CDS}$ for $WS_{10m} > 15$ m/s and $< 20$ m/s

**Figure 10: This figure shows mean values of ERA5 Charnock coefficient for strong wind speed conditions, and four different wind directional bins. The filters have been applied for each ERA5 node, that is: the timestamps selected for computing the Charnock coefficient values are not necessarily concurrent between all the nodes.**

### 3.3 Bias correction

From the above, we conclude that CFSv2 and ERA5 can come to a closer agreement when changing the value of the ERA5 Charnock parameter. This is demonstrated in Fig. 11, where ERA5 $\alpha_{Ch}$ values are set to 0.018, and 10 m ERA5 wind speeds are derived as explained in the previous section. The value of 0.018 has been chosen empirically by the authors of the report, from experience and preparatory works[7]. Other values reported in the literature include: 0.012 in (Araújo da Silva, 2022), 0.012 to 0.014 in (Peña et al., 2008b), 0.018 for the IFS ran without the ocean model[8] and in (Brown et al., 2013). Please note that the MOST should be used to obtain satisfactory results for small to medium wind speeds where atmospheric stability is important, i.e. using the Obukhov length as explained in Sect. 3.1.

---

[7] Published on https://eo-winds.net/2021/09/06/reconciling-surface-layers-wind-speeds-in-cfs-and-era5-reanalyses-lifehack/.
[8] See https://codes.ecmwf.int/grib/param-db/148

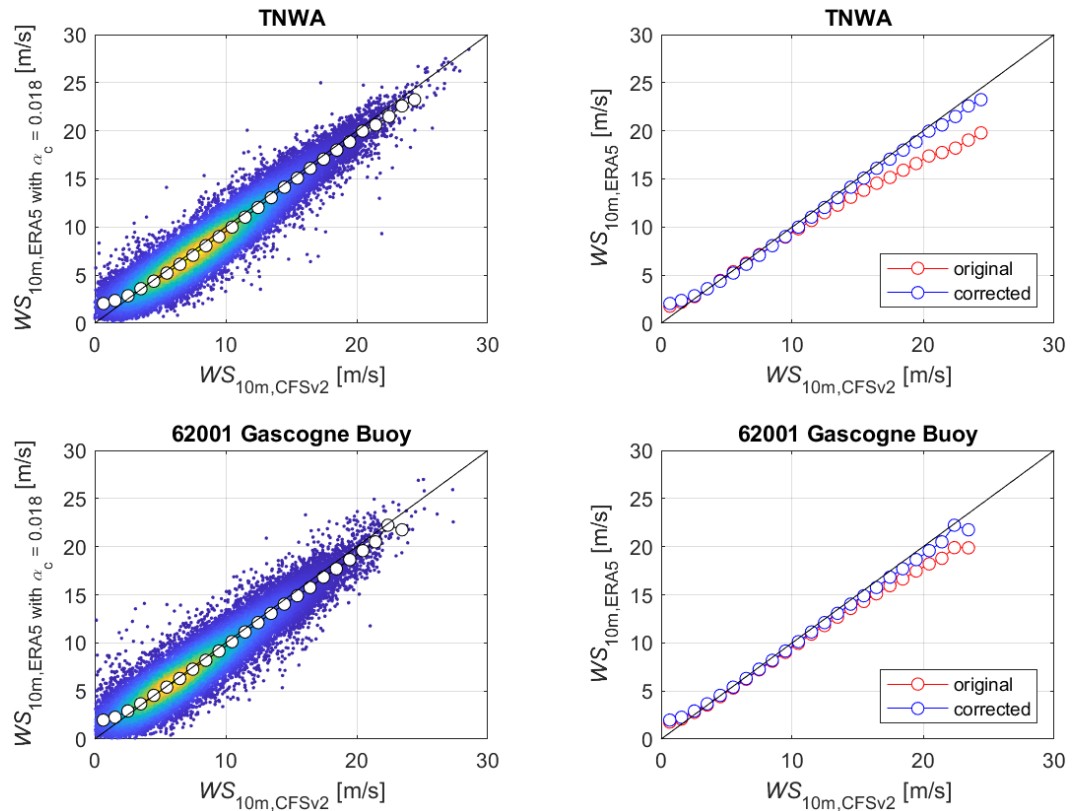

**Figure 11: This figure shows comparison of ERA5 10m wind speeds computed, as explained the text, using the CDS friction velocity and Charnock parameter capped at 0.018 and the CFSv2 time series. The white dots at the binned mean values (for bins with more than 10 points). This method reduced greatly the difference at large wind speeds.**

Furthermore, this methods allows a direct comparison of ERA5 values with measurements, without having to extrapolate the measurement elevation: see the example of the 26.1m anemometer time series at the IJmuiden mast in Fig. 12, where four time series are compared: original ERA5 (top-left), ERA5 with constant $\alpha_{Ch} = 0.018$ (top-right), ERA5 with $\max(\alpha_{Ch}) = 0.018$ (bottom-left) and the method employed in the Global Atlas of Siting Parameters Ocean and Coast (GASPOC) project (DHI, 2023) for their "*modified*" ERA5 wind speed time series (bottom-right). This last method uses ERA5 wind speed and atmospheric stability information only (not $\alpha_{Ch}$ nor $u_{*0}$), and it leads therefore to larger wind speeds in strong wind conditions. The method where $\alpha_{Ch}$ is capped to 0.018 is analogous to reduced drag methods is commonly used for ocean surface layer modelling, see for instance section 2.6 of (DHI, 2017) or the SWAN model documentation[9]. As shown in Table 2, the suggested correction method (constant $\alpha_{Ch} = 0.018$, or $\max(\alpha_{Ch}) = 0.018$) leads to a slight overestimation of strong wind speeds, while the original ERA5 data show an underestimation of almost 10% for the largest values.

---

[9] See https://swanmodel.sourceforge.io/online_doc/swantech/node15.html, accessed 2024-05-02.

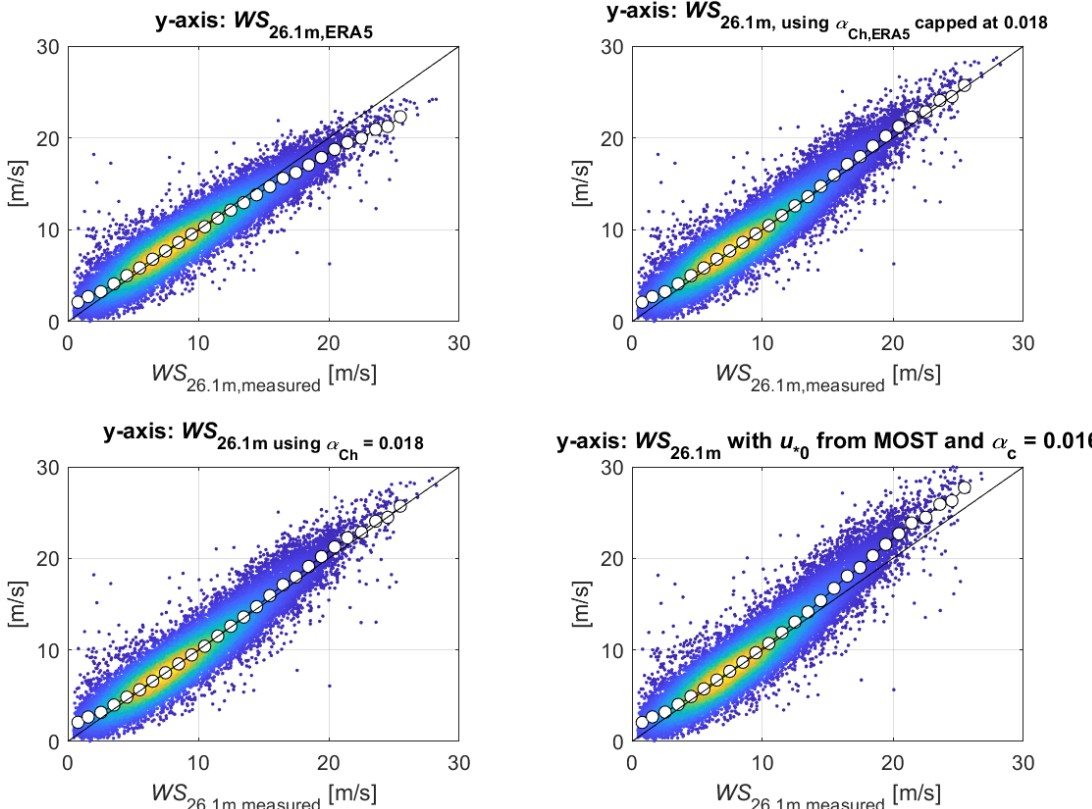

**Figure 12: This figure shows comparison of ERA5 26.1m wind speeds and measurements at 26.1 m from the Ijmuiden mast. The model data are computed as explained the text and summarized in the title of the subplots. The white dots at the binned mean values (for bins with more than 10 points).**

**Table 2: Pairwise linear correlation coefficient _R_ and mean relative differences between the four model time series described in the text and IJM hourly wind speed measurements at IJM (26.1 mMSL). The mean relative differences are computed for three different threshold of measured wind speeds.**

| Model time series | _R_ [-] | Mean relative differences [%] | | |
|---|---|---|---|---|
| | | WS > 10 m/s | WS > 15 m/s | WS > 20 m/s |
| ERA5 from CDS | 0.944 | -3.7 | -6.6 | -9.7 |
| ERA5 with constant $\alpha_{Ch} = 0.018$ | 0.947 | 2.2 | 4.3 | 4.1 |
| ERA5 with max($\alpha_{Ch}$) = 0.018 | 0.948 | 2.1 | 4.3 | 4.1 |
| "Modified" ERA5 wind speed time series from (DHI, 2023) | 0.945 | 5.3 | 9.2 | 10.3 |

### 3.4 Limitations and applicability of the proposed methodology

The suggested correction method presented earlier is not free from limitations. In this section, we discuss four main topics which should be investigated further: a) the choice of the Charnock parameter, b) the validity of the method for wind speeds larger than 25 m/s, c) the derivation of wind speed values for shorter averaging period than 1-hour, and d) validation for long fetches.

As explained in Sect. 1.1, all the drag formulations presented in this study require some degree of tuning using one or more empirical parameters. In our case, we propose a single value of the Charnock parameter, 0.018, which gives satisfactory results for correcting ERA5 wind speeds so that they match CFSR/CFSv2 values (the industry preferred model, see Sect. 1) as well as measurements. For a given offshore wind project, the practitioner may change or adapt this value to best fit their objectives, and the choice of this parameter should always be documented, discussed, and contextualised.

Our measurement datasets do not contain 10 m hourly wind speeds larger than 25 m/s. For tropical cyclone conditions where 10 m wind speeds can exceed 30 m/s, judging from the results presented in (Janssen and Bidlot, 2023) and (Bouin et al., 2024) about drag coefficient reduction at such large wind speeds, the proposed method likely overestimates measured values.

For engineering applications, it is often necessary to assess extreme values for averaging times shorter than 1-hour, typically 10-minutes, 1-minute and 3-second. The IFS and GFS NWP modelling systems do not model explicitly microscale turbulent eddies, and therefore rely on parametrisations; see Section 3.10.4 of (ECMWF, 2016a). In engineering, measurement-based methods are used (Andersen and Løvseth, 2006), alongside with model-based, empirical spectral correction methods (Larsén and Ott, 2022). Our correction method only deals with 1-hour wind speeds, for deriving shorter averaging we refer to the two above-mentioned studies.

Finally, our study does not show a validation for long fetch conditions, where, as shown in Fig. 5, differences between CFSR/CFSv2 and ERA5 are smaller. This is because the measurement datasets currently publicly available are primarily in enclosed seas, or in places where the strongest 10m winds come from shore. There exist high-quality LiDAR measurements in open ocean, but they are not publicly available. When such datasets become publicly available, or for projects with such data, additional validations should be carried out.

Overall, it important to stress that our study does not conclude that using an uncoupled version of IFS, with a constant Charnock parameter, is preferable to using a coupled model. In effect, there are many advantages to using a coupled model as it reflects more truly the nature of air-sea interactions. Yet, for specific applications and regions, biases can appear, and we wish to demonstrate that, with all the necessary information, simple and transparent correction methods can be derived.

### 4 Conclusion and suggestions for future work

The characterisation of the interactions between wind, sea state and currents remains an active field of research. Flux measurements experiments have been carried out at a few locations only, and the theoretical basis for understanding these interactions is still developing (Ayet and Chapron, 2022), (Janssen and Bidlot, 2023).

Modellers have adapted these models to one- or two-way coupled atmosphere-ocean modelling systems, and comparisons are
regularly performed, see for instance (Edson et al, 2013) and (Bouin et al,, 2024). These comparisons are often carried out for
locations that are not representative of Offshore Wind locations. For instance, the ECWMF drag formulation in (Edson et al.,
2013) is representative of the "*globally averaged wave age–dependent roughness at a given wind speed*". As oceans cover
70% of the globe, these model grid cells are in their vast majority deep water, very far offshore locations where the fetch- and
bathymetry dependent bias discussed in Sect. 3 is less visible. Or, in (Bouin et al,, 2024) and (Janssen and Bidlot, 2023), the
focus is set on very large wind speeds during Tropical Cyclones (TC); this is a valuable for the forecasting community but
offshore wind applications require the entire range of wind speeds to be modelled accurately.

Some may argue that when focusing on modelling sea state hydrodynamic conditions, adjustments to the wind forcing are
equally important as the other wave spectra sources and sinks of energy (bottom friction, wave-wave interactions, etc). This is
true, but the end-users and in particular offshore wind practitioners, require both wind and waves/hydrodynamics to be correct.
As the next generation of high resolution (towards km-scale) reanalysis datasets are being prepared for production (ERA6/7,
MERRA-3) planning for sector- or class of user-specific validations becomes necessary, for instance and with increasing
complexity:

> ➢ Validation runs against mast, LiDAR and FLS measurements during a great number of storm events (this is commonly
>   done for commercial projects). Adding such datasets alongside legacy measurements such as met buoys, would prove
>   very valuable and persuasive towards wind energy practitioners. On the other hand, oceanographers who are relying
>   in legacy measurement would find an alternative, higher quality datasets to work with.
> ➢ Refined grid meshes in coastal areas where offshore wind projects are being developed and adapting model data
>   delivery to the needs of offshore wind practitioners.
> ➢ Including LiDAR measurements in air-sea interaction measurement campaigns (as in the WFIP3 project[10]).
> ➢ Continuing working towards unifying oceanographic and ABL meteorology frameworks, from a wind energy
>   perspective as discussed in (Shaw et al. 2022).

---

[10] See https://www.psl.noaa.gov/renewable_energy/wfip3/.

**Appendix A**

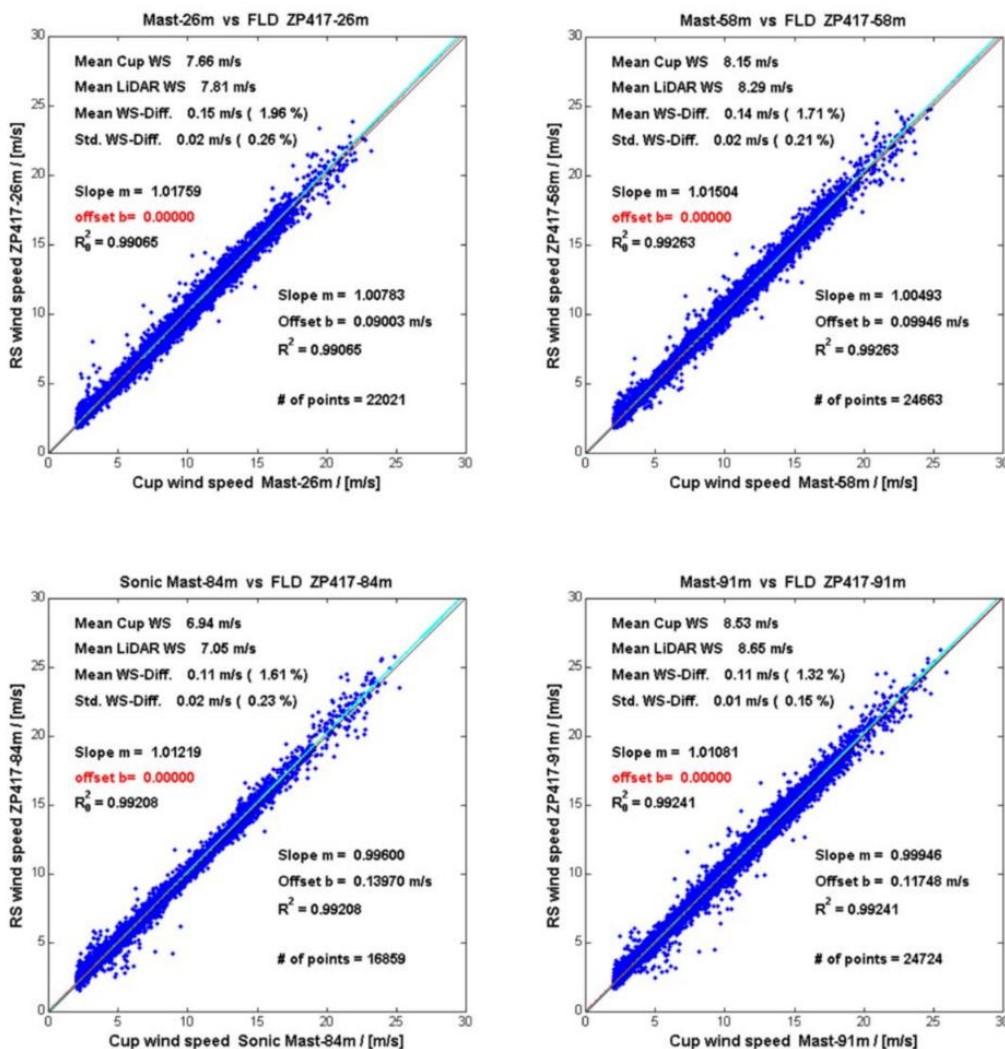

**Figure A1: Example of FLS LiDAR validation, from (GLGH, 2015)**

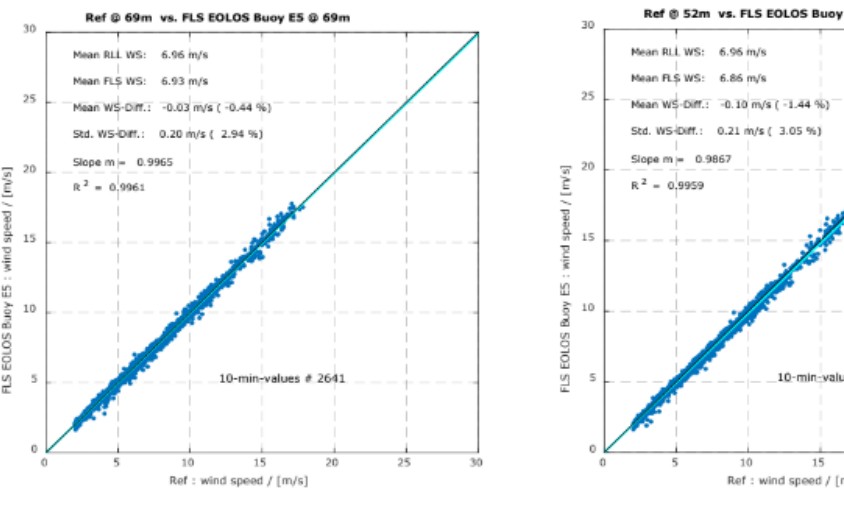

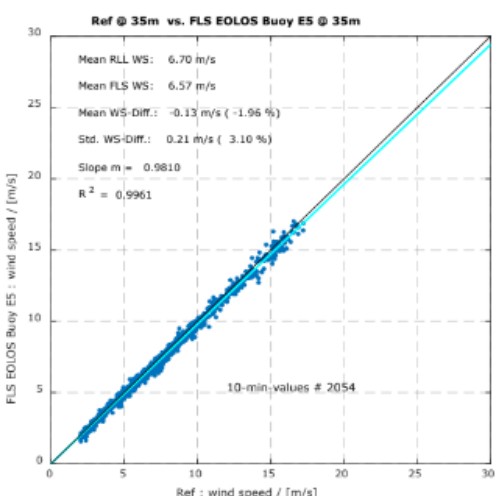

**Figure A2: Example of FLS validation, from (DNV GL, 2019)**

Lot 2: $WS_{100} \in [20; 30]$ m/s; $WD \in [210; 330]$ °N; $n_{samples}$ = 194; $\Delta\theta \in [-2;0.5[$°C

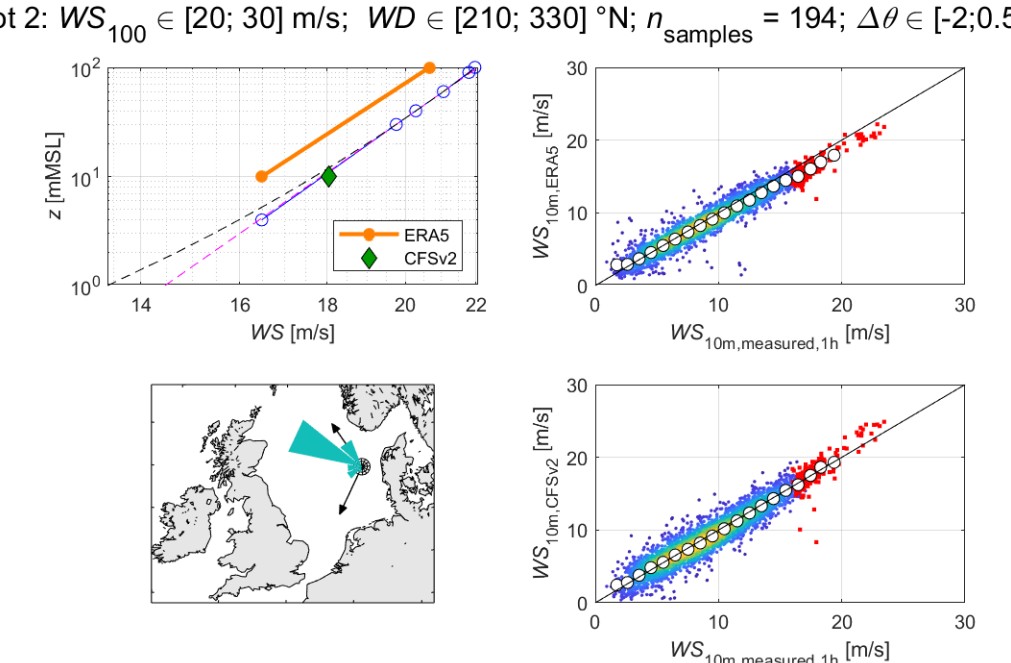

**Figure A3: Same as Fig. 6Figure 6, but for the Lot2 FLS location in the Danish North Sea.**

TNWA: $WS_{100} \in [20; 30]$ m/s; $WD \in [180; 20]$ °N; $n_{samples}$ = 59; $\Delta\theta \in [-2;0.5[$°C

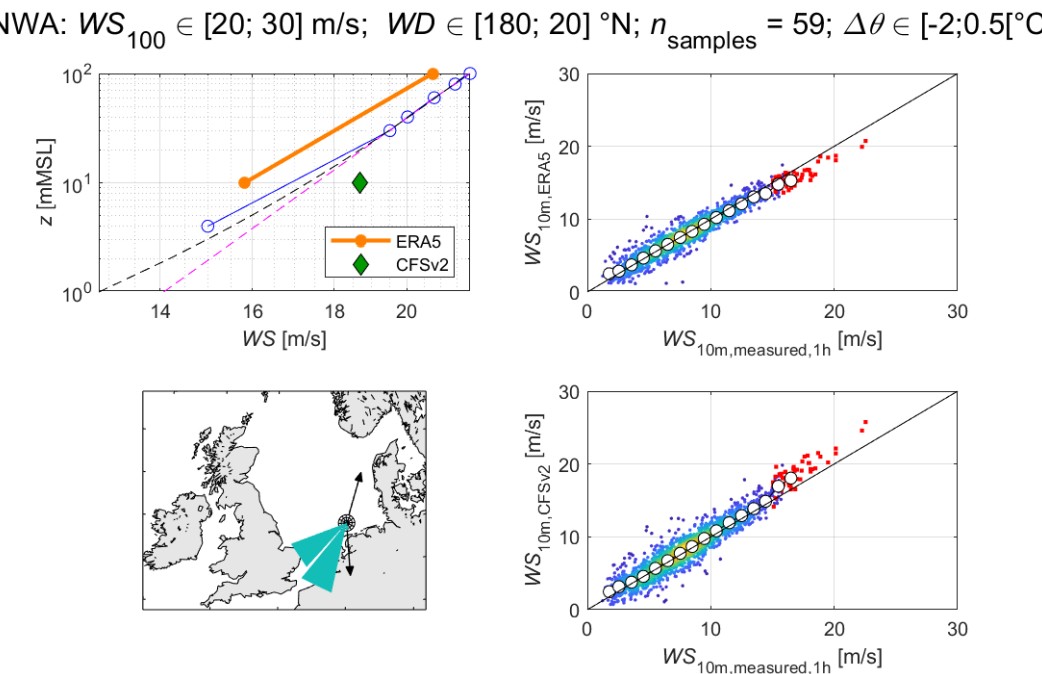

**Figure A4: Same as Fig. 6, but for the TNWA location in the Danish North Sea.**

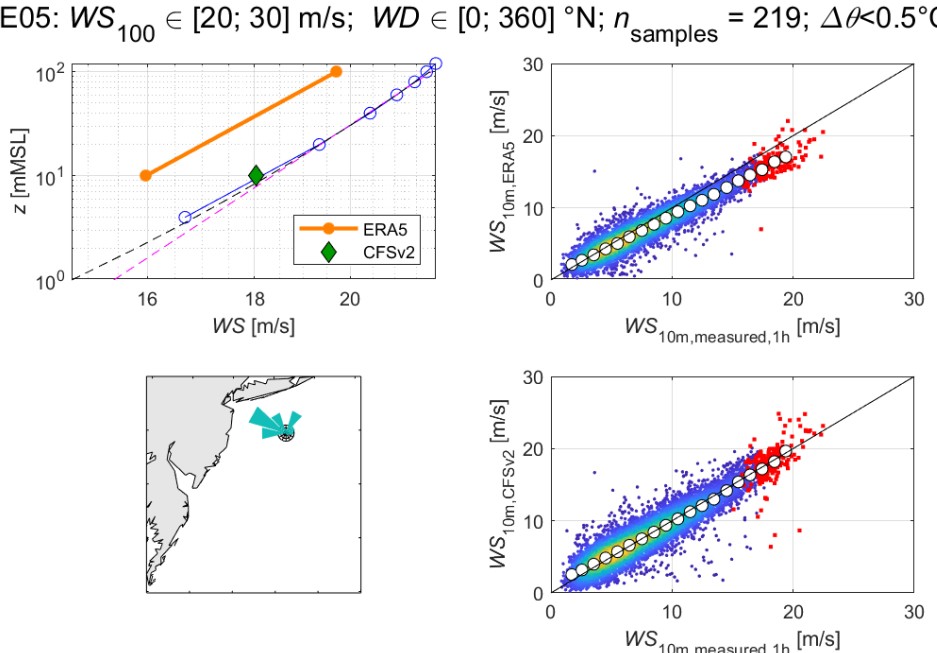


**Figure A5: Same as Fig. 6, but for the E05 FLS location in the New York Bight.**

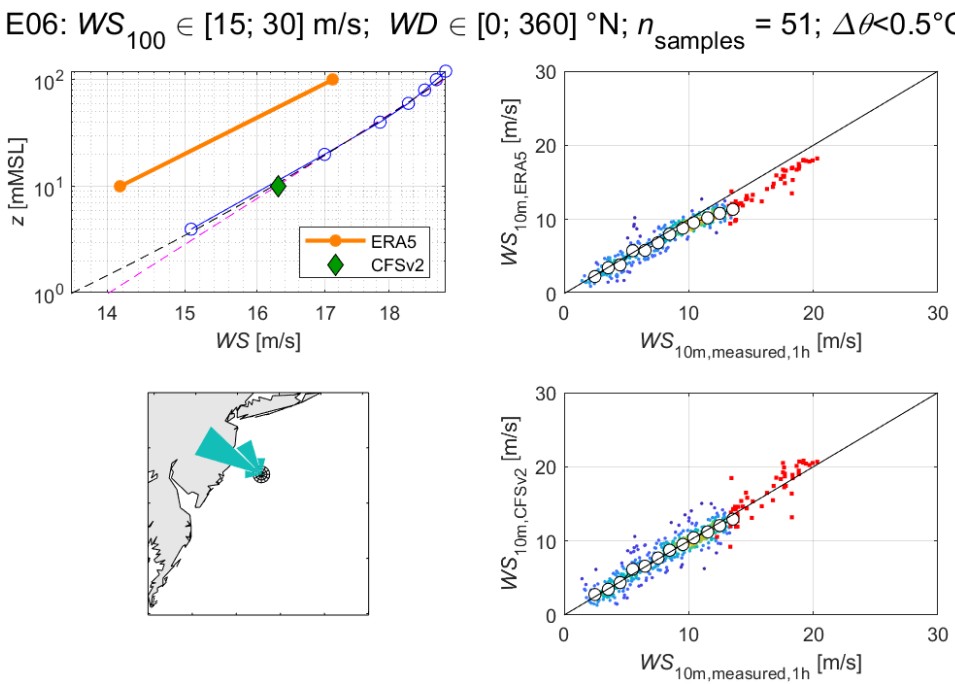

**Figure A6: Same as Fig. 6, but for the E06 FLS location in the New York Bight.**

## Code and data availability

The code is not available, as some of the functions are proprietary to C2Wind ApS. The data are publicly available, see the references in the article.

## Author contribution

RG conceptualised the article and carried out the analysis. JG reviewed the manuscript.

## Competing interests

The authors declare that they have no conflict of interest.

## Acknowledgements

The authors would like thank Jean Bidlot for his patience and his interest in Offshore Wind applications.

## Supplementary material

Publicly available LiDAR and FLS validation reports, as well as comparisons of ERA5 and CFSR/CFSv2 10 m wind speeds as several locations are provided in the following Zenodo repository.

Gandoin, R. Supplementary material to the Wind Energy Science article "Underestimation of strong wind speeds offshore in ERA5: evidence, discussion, and correction". Zenodo. Version v2. https://zenodo.org/records/11100768, 2024.

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
