# Peer review of "Underestimation of strong wind speeds offshore in ERA5: evidence, discussion, and correction"

_Wind Energy Science, 2024_

## Referee Comment (RC1)

**Review Comment**

April 2, 2024

Title: Underestimation of strong wind speeds offshore in ERA5: evidence, discussion, and correction

Author(s): Rémi Gandoin and Jorge Garza

MS No.: wes-2024-27

MS type: Research article

Iteration: Initial submission

**General comment**

This manuscript is discussed about the reason of underestimation for strong wind speed of ERA 5 and proposed its correction method. The reason of underestimation is clearly identified by comparison with other reanalysis data and some modeling parameter. Also, the efficiency of correction method is validated with trustable in situ measurement data. Although there is lack of discussion, the method is clearly mentioned and methodology is useful.

In conclusion, this manuscript has some scientific interest and is important for the industry, a reviewer would like to accept with following some revisions.

**Specific comment**

| Clause/ Subclause | Line number | Comments |
|---|---|---|
| 1. | 24-29 | The authors need to explain more detail about each paper referred here. Also, there is no explanation about correction in (DHI, 2023) refence web page. |
| | 34 | "the methods published so far only partially address the bias"
The authors need to add reference(s) mentioned here. |
| 1.1 | 55, eq.(1) | (Peña et. al, 2008) and many other papers define eq.(1) as "$-\psi_m(z/L)$". Although it depends on how the authors define "$\psi_m(z/L)$", it is suggested to use "$-\psi_m(z/L)$", as long as the authors refer (Peña et. al, 2008) without special note. |
| | 65 | Numerical value for $\alpha_M$ is not appeared in this paper. These parameters are important to calculate $z_0$. |
| | 67 | Correct "(GSF)" to (GFS). |
| | 73 | "$\tau$ is the wind stress ($u*_0^2$)" |

| | | |
|---|---|---|
| | | $\tau$ is not equal to $u*_0{}^2$ but equal to $\rho_a u*_0{}^2$. |
| | 74 | "Drag neutral coefficient" is may not commonly used (sounds like drag is 0). It is recommended to correct the word to "neutral drag coefficient" or "drag coefficient under neutral condition". |
| | 75 | Is "$C_d{}^2$" in Eq.(4) typo of $C_{d,n}{}^2$? |
| | 76 | It is recommended to explain that Un is wind speed for neutral condition. |
| 1.2 | 78 | "wind energy measurement datasets" sounds like energy(power) related measurement data. It is recommended to change it to "wind measurement datasets (created/used by wind energy industry, or add specific project name(s), reference(s) etc.)." |
| | Table 1 | It is recommended to add measurement height used in this study in this table, at least. |
| | Figure 2 | Geographical location of left bottom box in the figure is difficult to understand. The authors need to Add explanation such as location name or highlight box etc. in the map. |
| 1.3 | 127 | "the range of air-sea temperature difference to $\Delta\theta = T_{4m} - SST < [-2; 0.5]$ °C (North Sea) and $\Delta\theta = T_{4m} - SST < 0.5$°C (Atlantic Bight)"
 It is hard to understand the reason that stability criteria in North sea and in Atlantic Bight are different. Although this part is less important for this paper, the authors need to briefly explain or show references. Also, "$< [-2; 0.5]$ °C" might be "$\in [-2; 0.5[$ °C" (inconsistency with figure 3 etc.). |
| | Figure 3 | Is "[-2;0.5[" semi-open or typo of "[-2;0.5]"?
 There are many abbreviations (e.g. WS100, WD, PL, LL). in the figure. The authors need to explain in figure title or main body. |
| 1.4 | 141, Title | "1.3 Model data"
 Duplicated sub clause number |
| | 142-145 | Model data is grided, so these are not located on exact position(longitude-latitude) of in-situ measurement. The authors should explain how these differences were |

| | | handled. (e.g. interpolated, used nearest grid etc.) |
|---|---|---|
| 2.1 | 155 | "the ERA5 model results show larger differences for short fetches"

 This expression is not good because CFSv2 is not true value. What we can recognize here is just relative relation between ERA5 and CFSv2 depends on fetches. I suggest to modify this to similar expression used for title of figure 5, line 166. |
| | Figure 4 | Explain the meaning and values of the colors using color bar etc. Also, it is needed that the explanation of the white circle plots. |
| | 161-163, title of figure 4 | CFSR data is not shown in this figure. It is recommended to remove explanation regarding CFSR data due to avoid confusion. |
| 3 | Title | .(dot) in the end of clause title is not needed. |
| 3.1 | Figure 7 | I recognized $WS_{10m,CDS}$ and $WS_{10m,MOST}$ are essentially derived from the same numerical model, both based on ERA5. However, the scatter between those data seems almost the same level of scatter as, for example, top-right of figure 6. The authors have to explain the possible reasons. |
| | Figure 10 | Wind speed/direction is not uniform in area displayed in figure. Explain the reference coordinate. |
| 3.3 | 234 | There is no detail expression that why the authors chose the value of $\alpha_{ch} = 0.018$. The authors have to explain the reason. |
| | 243-234 | "ERA5 with $max(\alpha_{ch}) = 0.018$ (bottom-left)"

 Cap method or reduced drag coefficient ($z0$ as well) for strong wind is commonly used for ocean surface layer modeling. The authors should show reference(s) caused to get this idea, if there is. |
| | Fig.11 | Quantitative evaluation is not done in this paper. The authors need to show general statistics such as bias, slope, correlation coefficient etc., at least in this figure (recommended to add for other figures). |
| 4 | | The authors need to discuss about generality, limitation, applicability etc. of proposed methodology. For example, |

| | | $\alpha_{ch}$ = 0.018 or this method is ERA5 specific? Applicable for extreme wind speed (validated wind speed is up to 25m/s, validated against 1-hour average wind speed but 10-min average is needed for extreme wind)? etc. |
| --- | --- | --- |
| | | Also, $\alpha_{ch}$ = 0.018 is originally used in non-coupled ECMWF IFS, so people may think that it better to use non-coupled IFS model directly. Showing separate validation results for long and shot fetch may be good to explain the advantage of present method. |

.

---

## Referee Comment (RC2)

**Underestimation of strong wind speeds offshore in ERA5: evidence, discussion, and correction**
*Gandoin and Garza*

**REVIEW**

**GENERAL COMMENTS**
1. The writing style is extremely terse, and while each writer has their own style, I would encourage to add a bit more context to the storyline throughout the paper, since it becomes sometimes hard to follow since very little details are given, especially about a detailed interpretations of the plots shown and their implication.
2. Is it correct the results only consider neutral and unstable conditions? If so, this should be highlighted way more in the paper, and a "neutral and unstable conditions" specification should be added every time the main results from the study are discussed, potentially including the title.
3. ERA-5 has data at heights that can be directly compared with lidar observations. Why not including such a direct comparison to confirm the validity of your results, without the need of wind speed vertical extrapolation?

**MINOR COMMENTS**
1. L. 29: please explain "for design, slightly conservative values are typically desirable" in more detail
2. Fig. 2: what do the values of 'landmask' for ERA5 mean? Please clarify why values are not either 0 or 1 as one might expect.
3. In section 1.3, please specify which variables, at which height(s) are downloaded/considered from the models.
4. L. 125: have you checked your statement that "wind speeds larger than 15 and 20 m/s (where stable conditions are very rare)" at all sites?
5. Figures: you need to define all symbols, colors, abbreviations shown in the figure, legend, and title. If not needed, remove them.
6. L. 155: please provide more context when you start making comparisons about fetch. What are you referring to, how did you segregate the data, etc.

7. There are several grammar errors throughout the manuscript. One example: in the Fig. 8 caption: "length values" not "lengths values". Please double check your grammar.
8. Figg. A1 and A2 are impossible to read – make all fonts larger.

TECHNICAL CORRECTIONS
1. L.16: do not capitalize "power"
2. L. 26-27, 29 and many more: parentheses not needed for these references
3. L. 66: "NWP" was already defined
4. L. 79: "FLS" was already defined
5. L. 119: the sentence is not grammatically correct
6. L. 133: comma missing after "i.e."
7. L. 143: typo in "MOoD"
8. L. 157: a verb is missing in this sentence.
9. Fig. 6: do we need all the info in the title? If so, please explain what they are referring to, as no information is included in the caption or text.
10. L. 195: "at" instead of "are"?
11. L. 212: "latter" not "later"
12. L. 213: "leads" not "lead", and "to a 60%" not "to 60%"
13. Copernicus requires you to list a DOI for all references that have one.
14. L. 297: "The analysis was carried out in MATLAB" is probably not needed since the code is not made availably anyways.

---

## Author Comment (AC1)

Title: Underestimation of strong wind speeds offshore in ERA5: evidence, discussion, and correction
Author(s): Rémi Gandoin and Jorge Garza

**Response to Review Comment 1: Anonymous Referee #1**

The original comments from Referee #1 can be found at https://wes.copernicus.org/preprints/wes-2024-27#RC1. They have been reproduced below. Answers from the authors are marked in blue.

**General comment**

This manuscript is discussed about the reason of underestimation for strong wind speed of ERA 5 and proposed its correction method. The reason of underestimation is clearly identified by comparison with other reanalysis data and some modeling parameter. Also, the efficiency of correction method is validated with trustable in situ measurement data. Although there is lack of discussion, the method is clearly mentioned and methodology is useful.

Many thanks for your comment and your thorough review.

**Specific comment**

Section 1 Lines 24-29: The authors need to explain more detail about each paper referred here. Also, there is no explanation about correction in (DHI, 2023) refence web page.

We have added a short explanation about the context of these articles. The reference (DHI, 2023) has been added to the supplementary material (a version 2 of the supplementary material has been created).

Section 1 Line 34: "The methods published so far only partially address the bias". The authors need to add reference(s) mentioned here.

We have added references to the studies (Alday et al., 2021) and (DHI, 2023).

Section 1.1 Line 55 Eq. (1): (Peña et. al, 2008) and many other papers define eq.(1) as "$-\psi_m(z/L)$". Although it depends on how the authors define "$\psi_m(z/L)$", it is suggested to use "$-\psi_m(z/L)$", as long as the authors refer (Peña et. al, 2008) without special note.

Thanks for this suggestion. We have applied for the changes.

Section 1.1 Line 65: Numerical value for $\alpha_M$ is not appeared in this paper. These parameters are important to calculate $z_0$.

We have added a reference to the IFS documentation, and the value of $\alpha_M$.

Section 1.1 Line 67: Correct "(GSF)" to (GFS).

Thank you, we have corrected this typo.

Section 1.1 Line 73: "$\tau$ is the wind stress ($u*02$)"$\tau$ is not equal to $u*02$ but equal to $\rho au*02$.

Thank you, we have corrected the expression.

Section 1.1 Line 74: "Drag neutral coefficient" is may not commonly used (sounds like drag is 0). It is recommended to correct the word to "neutral drag coefficient" or "drag coefficient under neutral condition".

Thank you, this has been corrected.

Section 1.1 Line 75: Is "$C_{d2}$" in Eq.(4) typo of $C_{d,n2}$?

Thank you, this has been corrected.

Section 1.1 Line 76: It is recommended to explain that $U_n$ is wind speed for neutral condition.

We have added "for neutral conditions".

Section 1.2 Line 78: "wind energy measurement datasets" sounds like energy(power) related measurement data. It is recommended to change it to "wind measurement datasets (created/used by wind energy industry, or add specific project name(s), reference(s) etc.)."

We have reformulated with "measurement datasets from the wind energy industry".

Section 1.2 Table 1:  It is recommended to add measurement height used in this study in this table, at least.

Thank you, of course this is a good suggestion. We have added a column.

Section 1.2 Figure 2: Geographical location of left bottom box in the figure is difficult to understand. The authors need to Add explanation such as location name or highlight box etc. in the map.

Thank you, of course this is a good suggestion. We have labels for the Mid-Atlantic Bight and New York.

Section 1.3 Line 127: "the range of air-sea temperature difference to $\Delta\theta = T_{4m} - SST < [-2; 0.5]$ °C (North Sea) and $\Delta\theta = T_{4m} - SST < 0.5$°C (Atlantic Bight)"It is hard to understand the reason that stability criteria in North sea and in Atlantic Bight are different. Although this part is less important for this paper, the authors need to briefly explain or show references. Also, "$< [-2; 0.5]$ °C" might be "$\in [-2; 0.5[$ °C (inconsistency with figure 3 etc.).

We have corrected the typo in the interval, and added "The reason for choosing two different ranges of temperature differences, is that in the Mid-Atlantic Bight strong wind conditions occur during winter for very unstable conditions".

Section 1.3 Figure 3: Is "$[-2;0.5[$" semi-open or typo of "$[-2;0.5]$"?There are many abbreviations (e.g. WS100, WD, PL, LL). in the figure. The authors need to explain in figure title or main body.

There was a typo in the interval stated in the text. We have added clarifications in the caption. We agreed these were missing.

Section 1.4. Line 141, Title: "1.3 Model data"Duplicated sub clause number

Thank you, this has been corrected.

Section 1.3 Lines 142-145: Model data is grided, so these are not located on exact position(longitude-latitude) of in-situ measurement. The authors should explain how these differences were handled. (e.g. interpolated, used nearest grid etc.)

Yes, this was missing. We have explained which method was used for which dataset. For IJM and TNWA we have interpolated data available. For the others, we have used the nearest node as this is the method used for extracting the data on DHI's Metocean on Demand platform.

Section 2.1 Line 155: "the ERA5 model results show larger differences for short fetches" This expression is not good because CFSv2 is not true value. What we can recognize here is just relative relation between ERA5 and CFSv2 depends on fetches. I suggest to modify this to similar expression used for title of figure 5, line 166.

You are right, this a "model to model" comparison. We have replaced by: "Figure 5 shows that at the 62001 buoy location and when separating the dataset between short- and long fetches, the relative difference between the models seems to be larger for short fetches."

Section 2.1 Figure 4: Explain the meaning and values of the colors using color bar etc. Also, it is needed that the explanation of the white circle plots.

It is not straightforward to add a colorbar for this plot, so we have added in the caption that "The density of the scatter plot using a colormap, from blue (low density, few points) to yellow (high density, many points)." About the markers: we have added that "The white dots at the binned mean values (for bins with more than 10 points)."

Section 2.1 Lines 161.163, title of Figure 4: CFSR data is not shown in this figure. It is recommended to remove explanation regarding CFSR data due to avoid confusion.

Yes, we concur. We have removed the statement from the caption.

Section 3 Title: (dot) in the end of clause title is not needed.

Thank you, this has been corrected.

Section 3.1 Figure 7: I recognized $WS_{10m,CDS}$ and $WS_{10m,MOST}$ are essentially derived from the same numerical model, both based on ERA5. However, the scatter between those data seems almost the same level of scatter as, for example, top-right of figure 6. The authors have to explain the possible reasons.

Yes, this is an important point which we hope will be fixed in ERA6. We have added: "As explained above, $u_{(*0)}$ is provided as a forecast value while the 10 m wind speed from the CDS comes from the analysis (ERA5 contains both analysis and forecast fields); also, the Obukhov length L is computed from other model fields, and does not necessary correspond, for every timestamp, to the value used in the IFS. These differences explain the scatter between the reconstructed 10m wind speeds and the ones from the CDS, on the second plot in Fig. 7."

Section 3.1 Figure 10: Wind speed/direction is not uniform in area displayed in figure. Explain the reference coordinate.

Thank you, we realize this explanation was missing. We have added in the caption: "The filters have been applied for each ERA5 node, that is: the timestamps selected for computing the Charnock coefficient values are not necessarily concurrent between all the nodes."

Section 3.3 Line 243: There is no detail expression that why the authors chose the value of $\alpha_{ch} = 0.018$. The authors have to explain the reason.

Yes, this was missing. We have added "The value of 0.018 has been chosen empirically by the authors of the report, from experience and preparatory works. Other values reported in the literature include: 0.012 in (Araújo da Silva, 2022), 0.012 to 0.014 in (Peña et al., 2008b), 0.018 for the IFS ran without the ocean model and in (Brown et al., 2013)." (see the two added footnotes)

Section 3.3 Line 243-244: "ERA5 with max($\alpha_{ch}$) = 0.018 (bottom-left)"Cap method or reduced drag coefficient (z0 as well) for strong wind is commonly used for ocean surface layer modeling. The authors should show reference(s) caused to get this idea, if there is.

We have added: "The method where α_Ch is capped to 0.018 is analogous to reduced drag methods is commonly used for ocean surface layer modelling, see for instance section 2.6 of (DHI, 2017) or the SWAN model documentation".

Section 3.3 Fig.11: Quantitative evaluation is not done in this paper. The authors need to show general statistics such as bias, slope, correlation coefficient etc., at least in this figure (recommended to add for other figures).

Yes, you are right. We have added a table, Table 2, where statistics are provided for the comparisons displayed in Figure 12, that is the most important one in this article from our perspective. The text now says "As shown in Table 2, the suggested correction method (constant $\alpha_{Ch} = 0.018$, or $\max(\alpha_{Ch}) = 0.018$) leads to a slight overestimation of strong wind speeds, while the original ERA5 data show an underestimation of almost 10% for the largest values.".

Section 4: The authors need to discuss about generality, limitation, applicability etc. of proposed methodology. For example$\alpha_{ch}$ = 0.018 or this method is ERA5 specific? Applicable for extreme wind speed (validated wind speed is up to 25m/s, validated against 1-hour average wind speed but 10-min average is needed for extreme wind)? etc.Also, $\alpha_{ch}$ = 0.018 is originally used in non-coupled ECMWF IFS, so people may think that it better to use non-coupled IFS model directly. Showing separate validation results for long and shot fetch may be good to explain the advantage of present method.

Thank you for this important comment. We have added a new section, 3.4, to answer each of your points. We hope this is satisfactory.

---

## Author Comment (AC2)

Title: Underestimation of strong wind speeds offshore in ERA5: evidence, discussion, and correction
Author(s): Rémi Gandoin and Jorge Garza

**Response to Review Comment 2: Anonymous Referee #2**

The original comments from Referee #2 can be found at https://wes.copernicus.org/preprints/wes-2024-27#RC2. They have been reproduced below. Answers from the authors are marked in blue.

**General comments**

The writing style is extremely terse, and while each writer has their own style, I would encourage to add a bit more context to the storyline throughout the paper, since it becomes sometimes hard to follow since very little details are given, especially about a detailed interpretations of the plots shown and their implication.

Thank you this is an important comment. We have added text to the abstract, which describes the situation and give some context:

> ➢ "In turn, these time series are used for assessing wind, water levels and wave conditions, and are thereby key inputs to design activities such as calculation of fatigue- and extreme loads, as well as platform elevations"
> ➢ "If left uncorrected, poses a design risk (large- and extreme wind, waves and water level conditions are underestimated)"

Is it correct the results only consider neutral and unstable conditions? If so, this should be highlighted way more in the paper, and a "neutral and unstable conditions" specification should be added every time the main results from the study are discussed, potentially including the title.

Dear reviewer, no, the method applies for all stability conditions, but as discussed further in response to your minor comment no. 4:

> a) Deriving a 10m from lidar data is slightly more uncertain in stable conditions.
> b) Strong 10 m wind conditions (>15 m/s) occur very rarely in stable conditions.

Therefore, for the validation we have chosen to focus on unstable- and neutral conditions. But, as explained in Section 3.1 and Figure 7, one needs to derive the Obukhov length $L$ (from ERA5 time series) for the method to work satisfactorily for all wind speed ranges.

We have added the following clarifications to the text in the article:

> ➢ In footnote 6: "An alternative method is described on the ECMWF user support website at https://confluence.ecmwf.int/display/CKB/ERA5%3A+How+to+calculate+Obukhov+Length".
> ➢ Section 3.3 first paragraph: "Please note that the MOST should be used to obtain satisfactory results for small to medium wind speeds where atmospheric stability is important, i.e. using the Obukhov length as explained in Sect. 3.1".

ERA-5 has data at heights that can be directly compared with lidar observations. Why not including such a direct comparison to confirm the validity of your results, without the need of wind speed vertical extrapolation?

We have used the single levels data available from the CDS, and these are the data DHI have used as well on their Metocean on Demand database. You are right, there are raw(er) model data available but the download time is much longer and we have not been able to access these data.

**Minor comments**

1) L. 29: please explain "for design, slightly conservative values are typically desirable" in more detail.

Thank you this a good suggestion. We have added "that is: model results that underestimate large wind speeds, and thereby also large waves, pose a design risk (of too small loads, and too low platform elevations)."

2) Fig. 2: what do the values of 'landmask' for ERA5 mean? Please clarify why values are not either 0 or 1 as one might expect.

Thank you for this suggestion, we have added "For ERA5, and for IFS in general, land/sea mask values range from 0 to 1 and indicate the fraction of land in the model cell" in the caption with a footnote to https://confluence.ecmwf.int/display/FUG/Section+2.1.3.1+Land-Sea+Mask#Section2.1.3.1LandSeaMask-Land-Seamask.

3) In section 1.3, please specify which variables, at which height(s) are downloaded/considered from the models.

Thank you for this suggestion. We have added the required information in this section (which is now called Section 1.4 – there was a typo in the numbering).

4) L. 125: have you checked your statement that "wind speeds larger than 15 and 20 m/s (where stable conditions are very rare)" at all sites

Yes, we did, both in the reanalysis datasets (see Figure 1 below, we have also checked ERA5) and in the literature (Figure 2 below for the North Sea, ). We have added:

➢ "this was checked from both reanalysis data time series but also the literature, see (Sathe et al., 2011) for the North Sea"
➢ Reference (Sathe et al., 2011)

[Figure]

*Figure 1: Using CFSR/CFSv2 data. Top: scatter plot of air-sea temperature differences against 10 m wind speeds for three representative measurement locations. Bottom: histograms of 10 m / L where L is the Obukhov length derived from the Bulk Richardson number formulation from (Peña et al., 2008a), where stable conditions are identified for 10/L>0.05, following (Sathe et al., 2011).*

[Figure]

**Figure 6.** Variation of atmospheric stability with respect to wind speed between 225 and 315°. vs, very stable; s, stable; nns, near-neutral stable; n, neutral; nnu, near-neutral unstable; u, unstable; vu, very unstable.

*Figure 2: Reproduced from (Sathe et al. 2011), this figure shows that for large wind speeds in the North Sea and Western wind directions, strong winds occur very rarely for stable conditions.*

5) Figures: you need to define all symbols, colors, abbreviations shown in the figure, legend, and title. If not needed, remove them.

Thank you, we have received the same comment from Reviewer#1, and have updated the caption where it was needed, see our response to their comment.

6) L. 155: please provide more context when you start making comparisons about fetch. What are you eferring to, how did you segregate the data, etc.

Thank you, this was missing indeed. We have added "In this example, short fetches are defined as wind directions where wind comes from land across the Bay of Biscay while wind direction oriented towards the Atlantic Ocean are considered long fetches, see Fig. 2"

7) There are several grammar errors throughout the manuscript. One example: in the Fig. 8 caption: "length values" not "lengths values". Please double check your grammar.

8) Figg. A1 and A2 are impossible to read – make all fonts larger.

We have enlarged the Figures, thanks for this comment.

**Technical corrections**

1) L.16: do not capitalize "power"

This has been corrected.

2) L. 26-27, 29 and many more: parentheses not needed for these references

This has been corrected.

3) L. 66: "NWP" was already defined

This has been corrected.

4) L. 79: "FLS" was already defined

This has been corrected.

5) L. 119: the sentence is not grammatically correct

Thanks, we have now corrected the sentence to "However, for practical reasons this often needs to be done".

6) L. 133: comma missing after "i.e.

This has been corrected.

7) L. 143: typo in "MOoD

This has been corrected.

8) L. 157: a verb is missing in this sentence.

Thank you, we have now corrected to "This effect is of the same magnitude (…)"

9) Fig. 6: do we need all the info in the title? If so, please explain what they are referring to, as no information is included in the caption or text

Thank you, see our answer to your minor comment 5) earlier.

10) L. 195: "at" instead of "are"?

This has been corrected.

11) L. 212: "latter" not "later

This has been corrected.

12) L. 213: "leads" not "lead", and "to a 60%" not "to 60%"

This has been corrected.

13) Copernicus requires you to list a DOI for all references that have one

Thank you, we have checked the list and found, indeed, a handful of missing doi.

14) L. 297: "The analysis was carried out in MATLAB" is probably not needed since the code is not made availably anyways

This statement has been removed.